# Characteristics of Citrate-Esterified Starch and Enzymatically Debranched Starch and Their Effects on Diabetic Mice

**DOI:** 10.3390/foods13101486

**Published:** 2024-05-10

**Authors:** Nannan Wang, Changhe Ding, Yingying Xie, Jun Meng, Xing Fan, Duoduo Fan, Haowei Wan, Zhengqiang Jiang

**Affiliations:** 1College of Food Science and Engineering, Henan University of Technology, Zhengzhou 450001, China; wangn@haut.edu.cn (N.W.);; 2Food Laboratory of Zhongyuan, Luohe 462300, China

**Keywords:** citrate-esterified starch, enzymatically debranched starch, glycolipid metabolism, intestinal flora, T2DM

## Abstract

Chickpea has significant benefits as an adjuvant treatment for type 2 diabetes mellitus (T2DM). The properties of chickpea resistant starches (RSs) and their abilities to reduce T2DM symptoms and control intestinal flora were investigated. The RS content in citrate-esterified starch (CCS; 74.18%) was greater than that in pullulanase-modified starch (enzymatically debranched starch (EDS); 38.87%). Compared with those of native chickpea starch, there were noticeable changes in the granular structure and morphology of the two modified starches. The CCS showed surface cracking and aggregation. The EDS particles exhibited irregular layered structures. The expansion force of the modified starches decreased. The CCS and EDS could successfully lower blood glucose, regulate lipid metabolism, lower the levels of total cholesterol (TC) and low-density lipoprotein cholesterol (LDL-C), reduce the expressions of interleukin-6 (IL-6) and interleuki n-10 (IL-10), and decrease diabetes-related liver damage. Moreover, the CCS and EDS altered the intestinal flora makeup in mice with T2DM. The abundance of Bacteroidota increased. Both types of chickpea RSs exhibited significant hypoglycaemic and hypolipidaemic effects, contributing to the reduction in inflammatory levels and the improvement in gut microbiota balance.

## 1. Introduction

Type 2 diabetes mellitus (T2DM) is a widespread chronic metabolic disease caused by insulin resistance and hypersecretion [1,2]. This disease results in unpredictable harm to the heart, brain, kidneys, and peripheral nerves [3]. The incidence of T2DM has increased worldwide due to poor dietary habits and lifestyles, such as high-calorie diets and excessive sugar intake [4]. Currently, patients with diabetes constitute approximately 10% of the global population and play a significant role in the continuous growth of healthcare system expenditures [5].

Chickpea (*Cicer arietinum* Linn), which belongs to the legume family, is an annual or perennial herbaceous plant [6]. Chickpea is rich in starch, protein, dietary fibre, flavonoids, and vitamins [7]. Adding chickpeas to one’s diet can have numerous health benefits, such as increasing satiety, improving weight management, and preventing metabolic syndromes such as type 2 diabetes [8]. Therefore, chickpeas are a beneficial food for people with high blood pressure and obesity.

The starch content of chickpea is approximately 40% to 60%, with resistant starch (RS) accounting for approximately 8.4% to 19.7% of the starch content [8]. RS, a novel dietary fibre, has recently attracted the attention of researchers [9]. RS can stabilise postprandial blood sugar and insulin levels in patients with diabetes [10]. Moreover, RS significantly regulates the expression of glucose, reduces cholesterol, and inhibits the accumulation of fat [11]. Compared to those in the high-fat group, chickpea RSs exhibited lipid-lowering effects in a dose-dependent manner [12]. Kudzu resistant starch (KRS) was shown to significantly reduce the levels of fasting blood glucose, total cholesterol (TC), total triglycerides (TGs), high-density lipoprotein cholesterol (HDL-C), and low-density lipoprotein cholesterol (LDL-C) in T2DM mice [13]. In addition, KRS alleviated the intestinal flora dysbiosis caused by T2DM. Compared to those in the model control group, the blood glucose, TC, and TG concentrations in corn RS-treated rats significantly decreased, while the concentration of HDL cholesterol increased [14].

Research on the preparation of RS has focused primarily on corn, kudzu, and rice [13,15], and there is a limited number of studies in the literature on the preparation of chickpea RS using the pressure-heating enzyme method. Moreover, the yield of RSs from chickpeas using this technique is low, and only high-fat mouse models have been studied [16,17]. In this study, citric acid esterification and enzymatic debranching methods were used to modify chickpea starch. High yields of RSs were obtained. The changes in the physicochemical properties and structural characteristics of the chickpea starch were investigated before and after modification. In addition, the effects of chickpea starch and its RS on glucolipid metabolism and the intestinal flora of T2DM mice were evaluated.

## 2. Materials and Methods

### 2.1. Materials

Chickpea was purchased from retail markets in Zhengzhou, China. Pullulanase was obtained from Shanghai Yuanye Biotechnology Co., Ltd. (Shanghai, China). All other chemicals were of analytical grade.

### 2.2. Preparation of Chickpea Starch

The preparation of chickpea starch was performed with slight modifications as described by Zhang et al. [18]. Chickpea (5 g) and 0.3% (*w*/*v*) Na_2_SO_3_ were mixed at 40 °C for 16 h. The starch was dried at 40 °C and then crushed using a 100-mesh sieve.

### 2.3. Preparation of Chickpea RS

#### 2.3.1. Preparation of Enzymatically Debranched Starch (EDS)

Chickpea starch (5 g) prepared as described in Section 2.2 was dispersed in phosphate buffer (45 mL, pH 4.4) to form a slurry. The slurry was boiled for 10 min and then cooled to 58 °C. Pullulanase (20 U) was added to the slurry. After incubation at 58 °C for 6 h, the mixture was subjected to 100 °C for 20 min, cooled, and stored at 4 °C for 24 h. After being dried at 45 °C, the mixture was crushed using a 100-mesh sieve (Zhejiang Shangyu Daoxuwusi Instrument Factory, Shaoxing, China).

#### 2.3.2. Preparation of Citrate-Esterified Starch (CCS)

Starch samples were processed according to the methods of Xie and Liu [19] with some modifications. Chickpea starch (3 g) prepared as described in Section 2.2 was mixed with citric acid (10 g) and dissolved in distilled water. Then, the pH of the solution was adjusted to 3.0 with 10 mol/L NaOH, and the solution was incubated for 16 h at room temperature. The dried samples were esterified at 120 °C for 3 h and then washed with absolute ethanol. The washed samples were dried at 45 °C and crushed using a 100-mesh sieve.

### 2.4. Determination of Starch Digestibility

The in vitro digestibility of chickpea starch was determined using the method of Englyst [20]. The percentages of RS, slowly digestible starch (SDS), and rapidly digestible starch (RDS) were determined.

### 2.5. Scanning Electron Microscopy (SEM)

The morphologies of the chickpea starch, EDS, and CCS were determined by SEM (Quanta FEG 250, FEI, Eindhoven, The Netherlands). The samples were fixed to an aluminium plate, coated with gold, and then photographed at an acceleration voltage of 10.0 kV at 1500× and 2000× magnifications.

### 2.6. Viscosity

The viscosities of the starch samples (3.0 g of starch in 25 mL of distilled water) were measured using a Rapid viscometer (RVA-Ezi, Polton Instrument Company, Sweden) according to the national standard GB/T 24852-2010 [21]. The measurements involved a change in tank temperature. The agitator was kept at 50 °C for 1 min, increased to 95 °C in 12 min, held there for 2.5 min, and then lowered to 50 °C in 12 min and held there for 1.4 min. The rotation speed was 960 r min^−1^ for the first 10 s and remained at 160 r min^−1^ afterwards.

### 2.7. Fourier-Transform Infrared (FTIR) Spectroscopy

FTIR spectra of starch samples were recorded with an FTIR spectrometer (Nicolet iS 10, Thermo Fisher Scientific, Waltham, MA, USA) at room temperature. A total of 1.0 mg of each sample was mixed with 100 mg of KBr, and the mixture was tableted. Spectra were recorded in the range of 400–4000 cm^−1^.

### 2.8. Expansion Force

The starch sample (0.1 g) was dissolved in water (10 mL) and incubated at different temperatures (55–95 °C) for 30 min. The starch suspension was cooled and centrifuged at 5000× *g* for 10 min. The supernatant was subsequently dried and weighed. The swelling power (SP) was calculated according to the following equation, where W is the weight of the dried supernatant (g):SP (g/g) = (0.1 − W) × (1 − W/0.1) × 100%/0.1

### 2.9. Animals and Treatments

Sixty male mice (C57BL/6J, 8 weeks old, 20–24 g weight) were purchased from Zhejiang Viton Lihua Laboratory Animal Technology Co., Ltd. All animal experiments were carried out according to the guidelines established by the Welfare and Ethics Review Committee of Zhengzhou University Laboratory Animal Center (approval number: ZZU-LAC 20211015[15]). The mice were housed under controlled humidity (40–70%) and temperature (20–26 °C) conditions throughout the experiment. After adaptive feeding with a standard basal diet for one week, the sixty mice were randomly divided into a normal feeding group (12 mice on normal chow) and a high-fat and high-sugar feeding group (48 mice on high-fat and high-sugar chow). The T2DM mouse model was established with a high-fat and high-sugar diet (Table 1). Streptozotocin (STZ) was intraperitoneally injected according to the method of Li et al. [22]. Forty-eight type 2 diabetic mice were randomly and equally divided into four groups as follows: (1) a high-fat and high-sugar group (DCN) that was given saline; (2) a chickpea starch group (CS) that was supplied with 234.5 g/kg/day CS; (3) a CCS group that was administered 234.5 g/kg/day CCS; and (4) an EDS group that was treated with 234.5 g/kg/day EDS. The mice were treated for 5 weeks (Table 1).

### 2.10. Analysis of Blood Glucose

Blood glucose was determined using a blood glucose meter (Yuwell Company, Zhenjiang, China).

### 2.11. Evaluation of Serum Lipid Metabolism and Inflammatory Factors

Blood was drawn from the eyes of the mice. After centrifugation at 3500× *g*, blood serum was collected. Enzyme-linked immunosorbent assay (ELISA) kits (Jiancheng Institute of Biological Engineering, Nanjing, China) were used to evaluate TG, TC, HDL-C, LDL-C, interleukin-6 (IL-6), interleukin-10 (IL-10), and tumour necrosis factor-α (TNF-α) levels.

### 2.12. Morphology of Liver Slices

The obtained fresh liver slices were immersed in 4% paraformaldehyde fixative for 24 h. After dehydration in an ethyl alcohol gradient series, the liver slices were embedded in paraffin and then sectioned at 4 μm. The cellular morphology of the liver slices was visualised with haematoxylin and eosin (HE) staining.

### 2.13. Analysis of Intestinal Flora

Biological information was analysed by amplifying the V4 and V5 regions in the contents of the mouse colon using 16S rDNA amplicon sequencing technology. The sequence of the upstream primer used for PCR was GTGCCAGCMGCCGCGGTAA, and the sequence of the downstream primer was CCGTCAATTCCTTTGAGTTT. Operational taxonomic unit (OTU) clustering and amplicon sequence variation (ASV) noise reduction were used to analyse the results. The obtained validation data were subsequently subjected to species annotation as well as abundance analysis to further explore the differences in community structure among the samples through alpha-diversity and beta-diversity analysis.

### 2.14. Statistical Analysis

The data were analysed using a one-way analysis of variance (ANOVA) with Tukey’s honestly significant difference (HSD) test at a significance level of *p* < 0.05 with SPSS 26.0. The data are presented as the means ± standard deviations.

A Venn diagram can be used to analyse common and unique feature sequences in sample groups. High-quality sequences were categorised into operational taxonomic units (OTUs) with a similarity truncation value of 98%. Alpha diversity was used to analyse the diversity of microbial communities in the samples. Alpha diversity indices, including Shannon and Simpson [23], were calculated using Quantitative Insights into Microbial Ecology (QIIME). The beta diversity among different groups was analysed using a principal coordinate analysis (PCoA). The PCoA results were displayed using the ade4 package and ggplot2 package in R software (Version 2.15.3). The unweighted pair group method with arithmetic mean analysis (UPGMA) is a common clustering analysis method used to solve classification problems. This method involves constructing a clustering tree to analyse the similarity among sample groups. Linear discriminant analysis (LDA) effect size (LEfSe) was used to find biomarkers with statistical differences between groups, which was presented by Histogram of Linear Discriminant Analysis distribution and cladogram [24]. The LEfSe software (Version 1.0) was used to conduct the LEfSe analysis (LDA score threshold: 4) so as to determine the biomarkers.

## 3. Results and Discussion

### 3.1. Digestive Properties of Modified Chickpea Starch

The results of the in vitro digestibility of the starch samples are shown in Table 2. The contents of RDS, SDS, and RS in the unmodified chickpea starch were 83.31%, 9.12%, and 7.57%, respectively. Compared to the unmodified chickpea starch, the EDS and CCS had lower RDS (58.06% and 25.16%), lower SDS (2.53% and 0.92%), and higher RS (38.87% and 74.18%) contents. Enzymatic debranching treatment and introducing chemical groups are effective ways to increase the content of RS. The content of RS in pea starch was shown to increase after debranching and regeneration using pullulanase [25]. Different concentrations of citric acid (5%, 10%, 20%, and 40% of the dry weight of starch) were used to esterify palm dry starch [26]. The SDS content increased from 31.71% to 39.43%. The content of RS increased from 37.55% to 53.38% [26]. These results are consistent with those of this study.

Pullulanase can effectively cleave starch to produce short-chain amylose. During the regeneration process, the formed linear starch molecules were rearranged through hydrogen bonding, resulting in a dense crystalline structure, enhancing resistance to hydrolysis by digestive enzymes, and increasing the content of RS [27]. During the citric acid treatment process, hydroxyl groups in starch molecules are replaced by citric acid groups, leading to the cross-linking of starch molecules, which resists enzymatic degradation and increases the RS content [28]. The increase in the RS content of citric acid-modified chickpea starch could be due to the entry of citrate molecules into the chickpea starch, increasing intermolecular cross-linking and reducing the impact of digestive enzymes on the starch chains.

### 3.2. Microstructure of Modified Starch

The microstructures of the starch particles modified by pullulanase and citric acid esterification debranching were observed through SEM (magnifications of 1500× and 3000×) to investigate the changes in the starch particles. The native chickpea starch particles appeared to be round or oval with a smooth and intact surface and did not display any apparent cracks (Figure 1). Compared with that of the native chickpea starch, the morphology of the modified chickpea starches exhibited varying degrees of disruption. The CCS displayed surface cracking and aggregation accompanied by the concavity of starch granule centres, resulting in a rounded configuration. Similar results have been found for tapioca starch [29] and potato starch [28]. In contrast, after enzymatic debranching, the starch granules were completely disrupted and presented an irregular layered structure with a rough adherent surface and starch aggregation. These results are consistent with the findings of an investigation on the enzymatic modification of waxy corn starch [27]. The reason for this might be the reassembly of linear starch into helical complexes during the regeneration process, increasing the density of the crystal structure [30].

### 3.3. Viscosity Properties of Modified Starch

The viscosity properties of native chickpea starch and modified chickpea starch are presented in Figure 2. Native chickpea starch gelatinised rapidly upon heating (Figure 2). The peak viscosity and the retention viscosity were 4326 cP and 3145 cP, respectively. The termination viscosity was 6176 cP. The decay and regeneration values were 1181 cp and 3031 cP, respectively. Compared to those of native chickpea starch, the viscosity curves of the citrate-esterified starch and enzyme debranched starch were approximately linear. The RVA curve of the EDS was slightly greater than that of the CCS. These results indicate that citrate esterification and enzymatic debranching prevent starch granules from gelatinising and swelling.

The peak viscosity of starch granules is associated with factors such as relative crystallinity, swelling power, and particle stability [31]. Citric acid esterification and enzymatic debranching reduced the swelling capacity and relative crystallinity of starch, leading to a subsequent decrease in peak viscosity [32]. It has been reported that the RVA curve of CCS is flatter than that of unmodified starch [28,33]. Treatment with enzymes reduced the pasting viscosity of native starch, which was similar to the findings of previous studies [16]. The RVA curves of the EDS were slightly greater than those of the CCS, possibly due to the small swelling of the molecules after water absorption.

### 3.4. Short-Range Orderliness of Modified Starch

The short-range ordered structures of chickpea starch and its RS were investigated using FTIR spectroscopy. Both the native and modified starches showed the characteristic absorption peaks of starch (Figure 3). The absorption peak at 3374 cm^−1^ corresponds to the vibration of starch hydroxyl groups (OH). The peak at 2936 cm^−1^ is attributed to C-H vibrations. The peak at 1641 cm^−1^ is attributed to C-O vibrations. The peaks at 1417 cm^−1^, 1160 cm^−1^, and 992 cm^−1^ are associated with -CH- vibrations, C-O bonds, and the vibration of each hydroxyl group, respectively.

Both CCS and EDS have major characteristic absorption peaks similar to those of RCS. After citric acid esterification, the characteristic peak at 1641 cm^−1^ was clearly weakened, and a new absorption peak attributed to C=O vibrations formed at 1739 cm-1. The presence of C=O bonds confirmed the successful introduction of a citric acid group into the chickpea starch molecule. Many starches, such as corn starch [34], wheat starch [35], and lentil starch [28], have also been shown to undergo ester bond formation during the synthesis of citrate starch esters. After enzymatic debranching, the amount of adsorbed hydrogen-bonded O-H groups in starch decreased, and its spectrum remained similar to that of native starch, indicating that the chemical groups of starch were not changed by enzymatic debranching. This indicates that more hydroxyl groups are involved in the formation of crystals during enzymatic debranching.

The wavenumbers 800 cm^−1^-1200 cm^−1^ were deconvoluted. The peaks at 1047 cm^−1^ and 1022 cm^−1^ represent the characteristic absorption peaks of the crystalline and amorphous regions of starch, respectively. The higher the FTIR absorbance ratio, the greater the degree of short-range order in starch. The ratio of 1047/1022 cm^−1^ for native chickpea starch was 0.69. After citric acid esterification, the ratio of 1047/1022 cm^−1^ decreased to 0.59. After enzymatic debranching, the ratio of 1047/1022 cm^−1^ increased to 0.73. This may be because citric acid esterification changed the crystalline structure of the starch, increasing the amorphous region and decreasing the internal order of the starch. In contrast, enzymatic debranching involves the recrystallisation of starch molecules. The short linear molecules formed through debranching reassemble with hydrogen bonding, creating dense crystalline structures and enhancing the short-range order of starch.

### 3.5. Expansion Force Analysis

The changes in the expansion force of chickpea starch and its native RS at different temperatures are shown in Figure 4. The expansion force of the chickpea starch was positively correlated with the temperature (55–95 °C) (Figure 4). There was no significant difference in the expansion force of the CCS at 55–65 °C compared with that of the RCS. However, the expansion force of the native chickpea starch was significantly greater than that of the citrate-esterified starch at temperatures ranging from 75 °C to 95 °C (*p* < 0.05). The same results were observed for potato starch [36] and lentil starch [28]. This is because the bonding interactions in the cross-linked structure formed by esterification reactions are greater than the hydrogen bonding forces in starch granules. Moreover, the substitution groups introduced by citrate esterification may alter the associations between different components within starch molecules, thereby limiting the expansion of starch particles [36,37]. Compared with the native chickpea starch, the EDS exhibited a greater expansion force at temperatures ranging from 55 °C to 75 °C (Figure 4). This phenomenon may be related to the structural changes in starch crystals during enzymatic debranching, accompanied by an increase in the linear starch content, as reported for sweet potato starch [38]. The expansion force of the CCS was significantly lower than that of the EDS (*p* < 0.05), which could be attributed to the higher bonding energy resulting from esterification than that of the double helix structure.

### 3.6. Effect of Modified Chickpea Starch on Blood Glucose in Mice

Before the intervention, the blood glucose levels of the mice in the model group and the intervention group ranged from 23 to 25 mmol/L with no significant difference between diabetic mice (Figure 5). After 5 weeks of dietary intervention, the blood glucose levels of the mice in the normal group exhibited relatively stable fluctuations. In contrast, the blood glucose levels of the mice in the experimental intervention group decreased significantly. These results indicate that chickpeas and their different RSs have a significant impact on the blood glucose levels of diabetic mice. The hypoglycaemic effect of CCS was the greatest, with no significant difference from that of EDS. However, the hypoglycaemic effect of RCS was the weakest. The reduction in mouse blood glucose levels is influenced by changes in mouse intestinal digestibility and hormone secretion. RS may regulate mouse intestinal metabolism to influence blood glucose levels [39].

### 3.7. Effect of Modified Chickpea Starch on Lipid Metabolism in Mice

T2DM is a complex chronic metabolic disease caused by insulin resistance in peripheral tissues [40]. Lipid metabolism can cause insulin resistance [41]. There is a close connection between lipid metabolism and glucose metabolism [42]. The inability to maintain lipid levels within the normal range over a long period is a major cause of chronic complications in patients with T2DM. Therefore, the regulation of lipid metabolism is an essential component of RCS modulation. TG, TC, HDL-C, and cholesterol LDL-C reflect the body’s lipid metabolism and possess many properties, such as anti-inflammatory, antioxidant, and antiapoptotic effects [43]. TC refers to the total amount of cholesterol in blood, reflecting the metabolism of body fat. TG is responsible for the storage and production of energy in the body. LDL-C is the transport carrier of endogenous cholesterol. HDL-C is a type of anti-atherosclerosis lipoprotein synthesised by the liver that facilitates reverse cholesterol transport to the liver for metabolism. It has been shown that T2DM is closely associated with elevated levels of TC and LDL-C [4].

The lipid metabolism of mice after 5 weeks of intervention is shown in Figure 6. Compared with those in the DCN group, the TC contents in the RCS group, CCS group, and EDS group significantly decreased (*p* < 0.05). The LDL-C content in the CCS group decreased significantly (*p* < 0.05) (Figure 6). The results indicate that CCS significantly regulated the lipid levels in the mice. These results demonstrate that providing diabetic mice with RS can significantly lower the levels of TC and LDL-C. It has been reported that RS supplementation can reduce the TC and LDL-C levels, enhance lipid metabolism, and lower blood glucose levels by promoting glycogen synthesis, inhibiting gluconeogenesis, and promoting lipid oxidation [44]. The intake of an appropriate amount of RS improves blood lipid abnormalities induced by a high-fat diet (HFD), which is driven by the restoration of the basal expression levels of transcription factors involved in lipid synthesis (SREBP-1c), cholesterol metabolism (SREBP-2), and fatty acid oxidation (PPARα) [45].

### 3.8. Effects of Different Modified Chickpea Starches on Inflammatory Factors in Mice

The inflammatory response is one of the main causes of T2DM pathogenesis and is primarily associated with changes in the levels of IL-6, TNF-α, and IL-10. The overexpression of these proinflammatory cytokines causes gluconeogenesis and insulin resistance. T2DM is associated with elevated levels of IL-6, IL-10, and TNF-α. Reducing the levels of IL-6, IL-10, and TNF-α helps to suppress inflammation and restore insulin receptor substrate activity [46]. The levels of inflammatory factors in the mice are shown in Figure 7. Compared with those in the DCN group, there were no significant changes in the levels of IL-6, IL-10, or TNF-α in the RCS group. Compared with those in the DCN group, the levels of IL-6, IL-10, and TNF-α in the T2DM group were considerably lower in the CCS intervention group (*p* < 0.05), but these levels were not significantly different from those in the CN group. The EDS intervention significantly reduced the IL-6 and IL-10 levels (*p* < 0.05), while the changes in the TNF-α levels were not significant. Thus, CCS and EDS can decrease the levels of IL-6 and IL-10 in diabetic mice.

### 3.9. Effect of Modified Chickpea Starch on Microstructure of Mouse Liver Tissue

The liver tissues of the mice in each group were stained and observed. As shown in Figure 8, the liver tissues of mice in the CN group were structurally intact, well-organised hepatocytes displaying a radial arrangement around the central vein. Moreover, intact nuclei were observed, and intracellular fat droplets were never found in liver tissues. In contrast, the liver lobules of the mice in the DCN group were disrupted with many lipid vacuoles and extensive hepatocyte degeneration. After the intervention, the damaged liver tissue structures in the DCN group, CCS group, and EDS group were restored, and the phenomenon of lipid vacuoles was significantly improved. The results of the CCS group and EDS group tended to be similar to those of the normal group, indicating that CCS and EDS significantly inhibited the development of the fatty liver.

### 3.10. Bioinformatic Analysis of Mouse Intestinal Flora

#### 3.10.1. Venn Diagram

There were 144 common core bacterial OTUs in the five samples, and the number of OTUs observed in the T2DM group was significantly lower than that in the normal group (Figure 9). In terms of the number of unique OTUs, the CN group had the most unique OTUs, followed by the EDS group, DCN group, and CCS group. The EDS was effective in increasing the species diversity of the mouse intestine. These results suggest that the use of different types of RS interventions will result in differences in intestinal flora diversity.

#### 3.10.2. Analysis of Differences between Groups

In terms of the Shannon index, the diversity of the CN group was significantly (*p* < 0.01) greater than that of the DCN group, RCS group, CCS group, and EDS group (Figure 10). After the intervention, the diversities of the RCS, CCS, and EDS groups were greater than that of the DCN group. Similarly, the Simpson indices of the RCS group and CCS group were significantly (*p* < 0.05) greater than that of the DCN group, and there was no significant difference between the intervention groups. The results indicate that the abundance and diversity of intestinal microorganisms in the T2DM group were significantly lower than those in the CN group. The microbial diversity improved after dietary interventions with RCS, CCS, and EDS. In terms of the alpha diversity analysis, there was no significant difference in microbial diversity among the RCS, CCS, and EDS intervention groups.

#### 3.10.3. Principal Coordinate Analysis (PCoA)

The DCN-EDS, CN-EDS, CN-CCS, and CN-RCS samples were distant from each other. There was an aggregation of sample points between the DCN group and the CCS group (Figure 11), which indicated that the intestinal flora structure of diabetic mice changed under the induction of a high-sugar and high-fat diet combined with STZ. The species composition of the mouse gut microbiota significantly differed after dietary intervention using RCS, CCS, or EDS.

#### 3.10.4. Unweighted Pair-Group Method with Arithmetic Mean Analysis (UPGMA)

The RCS group was the farthest from the CCS group, followed by the CN group (Figure 12). The distances between the CN group and CCS group were the closest. This indicates that there were significant differences in species composition among the RCS group, the CCS group, and the CN group. The species compositions of the DCN group, CN group, and EDS group were relatively similar. The CN group and CCS group had the most similar species structures. There was no significant effect on the structural aspects of species composition with CCS intervention. In contrast, dietary intervention using RCS and EDS had a greater effect on the structural aspects of the species composition.

#### 3.10.5. Linear Discriminant Analysis Effect Size (LEfSe) Analysis

The differentially abundant species in the CN group were *Prevotellaceae*, *Rikenellaceae*, *Oscillospiraceae*, *AlloPrevotella*, and *Alistipes* (Figure 13). The differentially abundant species in the DCN group were *Lachnospiraceae*, *Dubosiella*, and *Akkermansiaceae*. In the RCS group, the differentially abundant species were *Firmicutes*, *Ruminococcaceae*, and *Oscillospiraceae*. The differentially abundant species in the CCS group were *Bacteroidota*, *Bacteroidaceae*, *Sutterellaceae*, *Bacteroides*, and *Parasutterella*. The differentially abundant species in the EDS group were *Verrucomicrobiota*, *Akkermansiaceae*, *Enterobacteriaceae*, *Akkermansia*, and *Muribaculaceae*.

*Firmicutes* and *Bacteroidota* play crucial roles in regulating body metabolism, improving intestinal immunity, and promoting the body’s resistance to inflammatory responses [47]. *Ruminococcaceae* are essential for stabilising the intestinal barrier, improving diarrhoea, and reducing the risk of colon cancer [48]. *Parasutterella* can stably colonise the host intestine, participate in cholesterol metabolism, and maintain bile acid homeostasis [49]. *Akkermansia* is a beneficial intestinal mucin-degrading bacterium with probiotic effects on obesity, diabetes, and cardiovascular diseases. Mucus-degrading bacteria are negatively associated with maintaining intestinal integrity [50]. Higher levels of Akkermansia were observed in the intestines of patients with T2DM than in those of healthy individuals. *Muribaculaceae* is beneficial for intestinal epithelial health, and this species tends to thrive and grow under a high-fibre diet [51]. There were different species between the intervention groups, suggesting that the RCS, CCS, and EDS interventions had certain differences in regulating the diversity and abundance of the intestinal microbiota, thereby affecting their efficacy in reducing blood sugar and lipids.

#### 3.10.6. Phylum-Level Species Analysis

The distribution of the intestinal microbiota in each group of samples at the phylum-level is shown in Figure 14A. The main bacterial groups in the samples at the phylum level were *Firmicutes*, *Verrucomicrobiota*, *Bacteroidota*, and *Proteobacteria*. The abundance of *Firmicutes* in the RCS group was significantly greater (*p* < 0.05) than that in the EDS group. Compared with that in the DCN group, the abundance of *Bacteroidota* in the CCS group and EDS group was significantly greater (*p* < 0.05), and there was no significant difference between the two groups. In terms of *Verrucomicrobiota*, the abundance in the EDS group was significantly greater than that in the RCS group (*p* < 0.01), and there was no significant difference among the DCN group, CN group, and CCS group (Figure 15). In addition, EDS increased the abundance of *Proteobacteria*.

*Firmicutes* and *Bacteroidota* account for 90% of the intestinal flora abundance, making them the dominant bacterial groups in the gut. The *Firmicutes*/*Bacteroidota* (F/B) ratio is an indicator of intestinal health, and a relatively low F/B ratio plays crucial roles in regulating immune system homeostasis, reducing inflammatory responses, and maintaining cardiovascular health [47,52]. Compared to that in the CN group, the F/B ratio significantly increased in the DCN group induced by a high-fat high-sugar diet combined with STZ. After the dietary intervention, the F/B ratio significantly decreased (*p* < 0.05) and was not significantly different from that in the CN group (Figure 14B). These results suggest that the ability of RS to enhance inflammatory resistance and ameliorate disorders of glucose and lipid metabolism is driven by modulating the levels of dominant bacteria in the intestinal microbiota of T2DM mice.

#### 3.10.7. A Species Structure Analysis at the Genus Level

The distribution of the species abundance of the samples at the genus level is shown in Figure 16. The top 10 species in terms of abundance were *Akkermansia*, *Dubosiella*, *Bacteroides*, *Faecalibaculum*, *Muribaculaceae*, *Lactobacillus*, *Alloprevotella*, *Parasutterella*, *Alistipes*, and *Parabacteroides*. Compared to those in the CN group, the abundances of intestinal flora in the DCN group, RCS group, CCS group, and EDS group changed significantly (Figure 17). The abundance of *Akkermansia* in the EDS group was significantly greater (*p* < 0.05) than that in the RCS group. *Bacteroides* and *Dubosiella* play important roles in regulating glycolipid metabolism, improving intestinal immunity, and promoting the body’s resistance to inflammatory reactions [47]. The abundance of *Bacteroidota* in the CCS group was significantly (*p* < 0.05) greater than that in the DCN group and EDS group. The abundance of *Bacteroidota* in the RCS group was significantly (*p* < 0.01) greater than that in the CN group. The abundance of *Dubosiella* decreased significantly in both the RCS group and EDS group. In terms of *Lactobacillus*, the EDS group had the highest abundance, which was significantly (*p* < 0.05) greater than that in the DCN group. The intake of *Lactobacillus* can reduce blood glucose, blood lipids, and body weight in T2DM mice [51]. The abundance of *Muribaculaceae* in the EDS group was significantly greater than that in the CCS (*p* < 0.01) and DCN groups (*p* < 0.05). These results indicate that the EDS intervention significantly increased the abundance of *Muribaculaceae* in the mice compared to the CCS intervention and was superior to CCS in enhancing intestinal epithelial health. Compared to the DCN group, the EDS intervention group had a significantly greater (*p* < 0.05) abundance of *Parasutterella*.

## 4. Conclusions

After enzymatic debranching and citrate esterification, the content of chickpea RS increased from 7.57% to 38.87% and 74.18%, respectively. The apparent morphology of the two modified starches was different. The CCS granules could still maintain their original morphologies, but the surfaces showed cleavage and agglomeration. The EDS particles showed irregular layered structures. Compared to those of RCS, the expansion forces of both the CCS and EDS were significantly lower. This indicates that the modification treatment will change the structural and physicochemical properties of the starch. RS can reduce hyperglycaemia, hyperlipidaemia, and chronic systemic inflammation, improving insulin resistance. RS intervention can significantly alter the abundance and structure of the gut microbiota in T2DM mice. Different types of RS exhibit distinct activities in the diversification of the gut microbiota. These results suggest that RS may be a potential functional food for the prevention and treatment of T2DM.

In this paper, the effect of modified chickpea RS on a diabetic mouse model was studied, but the effect on patients with diabetes needs further study. In addition, further studies are needed to determine the amount of its addition in staple foods, such as bread, steamed bread, and noodles, not only to meet its functional properties, such as reducing postprandial blood sugar and improving intestinal flora, but also to consider its impact on the sensory quality of food. It is believed that these studies will provide theoretical guidance for the application of chickpea RS in the development of special food for the diabetic population.

## Figures and Tables

**Figure 1 foods-13-01486-f001:**
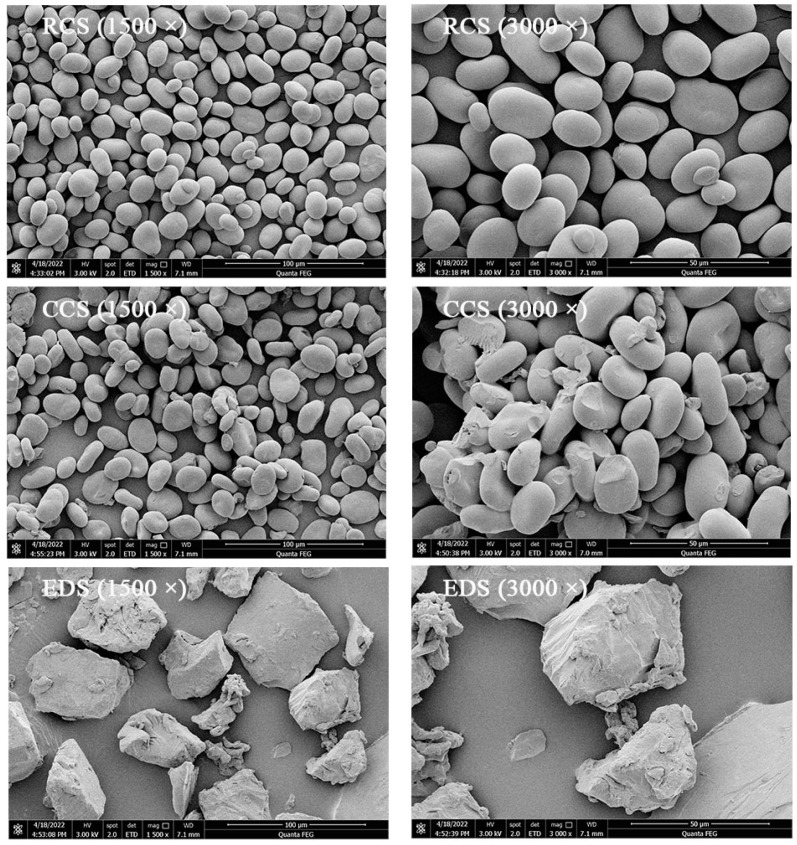
Scanning electron micrographs of modified chickpea starch. RCS, CCS, and EDS denote chickpea starch, citrate-esterified starch, and enzymatically debranched starch, respectively.

**Figure 2 foods-13-01486-f002:**
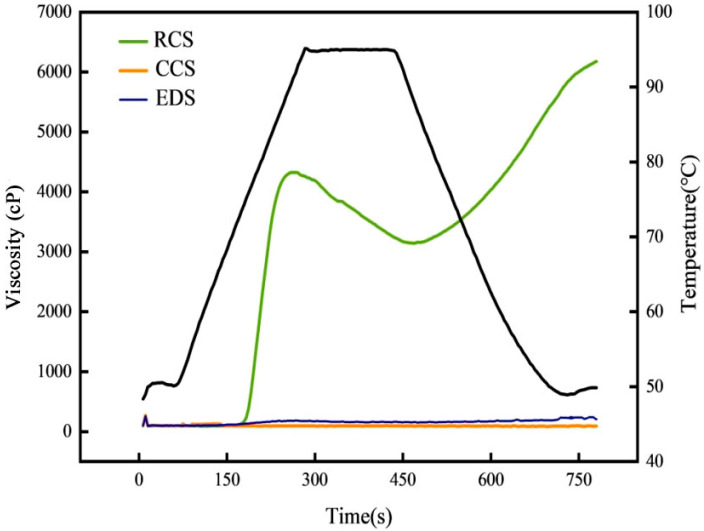
Viscosity curve of modified chickpea starch. RCS, CCS, and EDS denote chickpea starch, citrate-esterified starch, and enzymatically debranched starch, respectively.

**Figure 3 foods-13-01486-f003:**
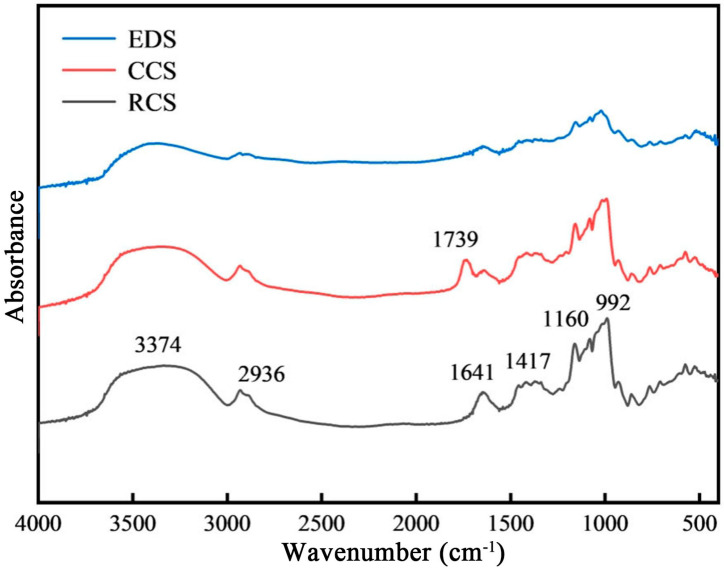
FTIR spectra of chickpea starch and its RS. RCS, CCS, and EDS denote chickpea starch, citrate-esterified starch, and enzymatically debranched starch, respectively.

**Figure 4 foods-13-01486-f004:**
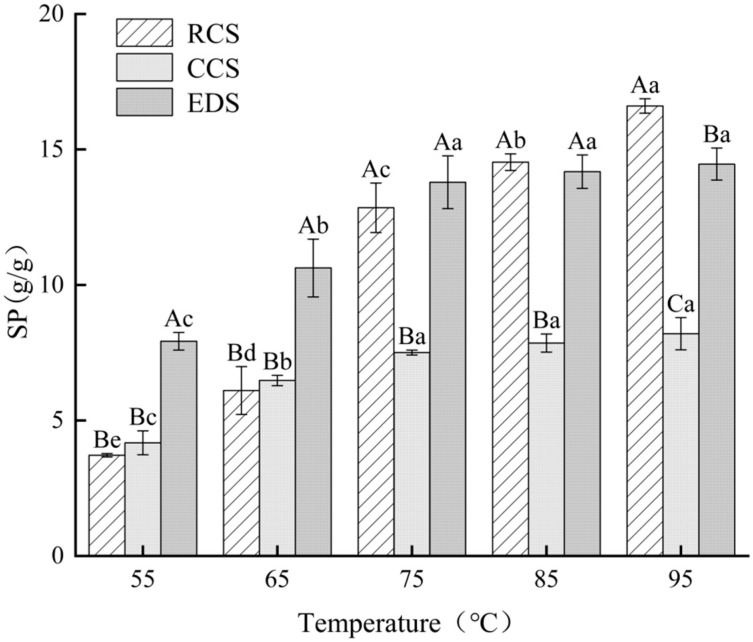
Expansion force of chickpea starch and modified chickpea starch. Each value represents mean ± SD. Different letters (a–e) indicate significant differences from each other at the different temperatures. Different letters (A–C) indicate significant differences from each other at the same temperature. Statistical analysis was performed using one-way ANOVA with Tukey’s post hoc test. RCS, CCS, and EDS denote chickpea starch, citrate-esterified starch, and enzymatically debranched starch, respectively.

**Figure 5 foods-13-01486-f005:**
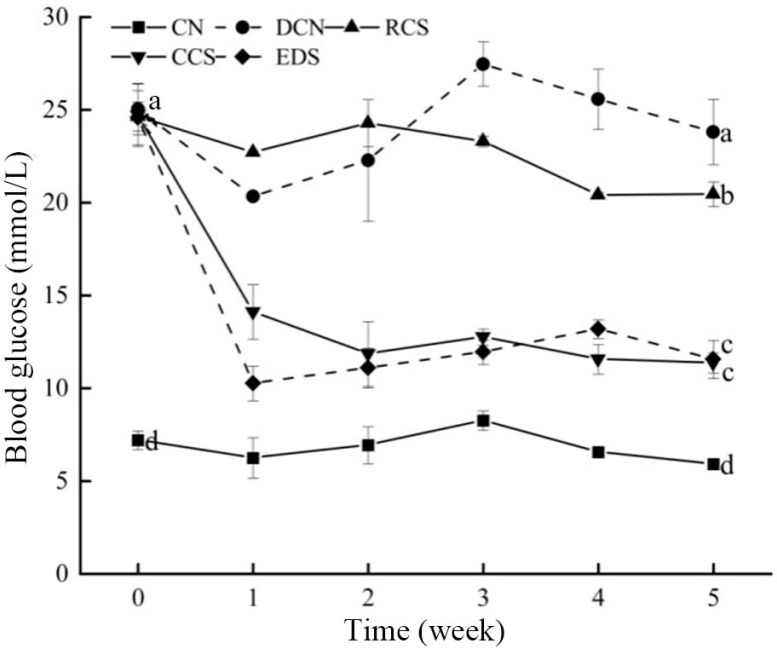
The concentrations of blood glucose in the mice. The blood glucose levels were measured weekly during the experimental period (5 weeks). The data are shown as the means ± SDs. Different letters (a–d) indicate significant differences from each other. A statistical analysis was performed using a one-way ANOVA with Tukey’s post hoc test. CN, DCN, RCS, CCS, and EDS denote the normal control group, high-fat and high-sugar control group, chickpea raw starch group, citrate-esterified starch group, and enzymatically debranched starch group, respectively.

**Figure 6 foods-13-01486-f006:**
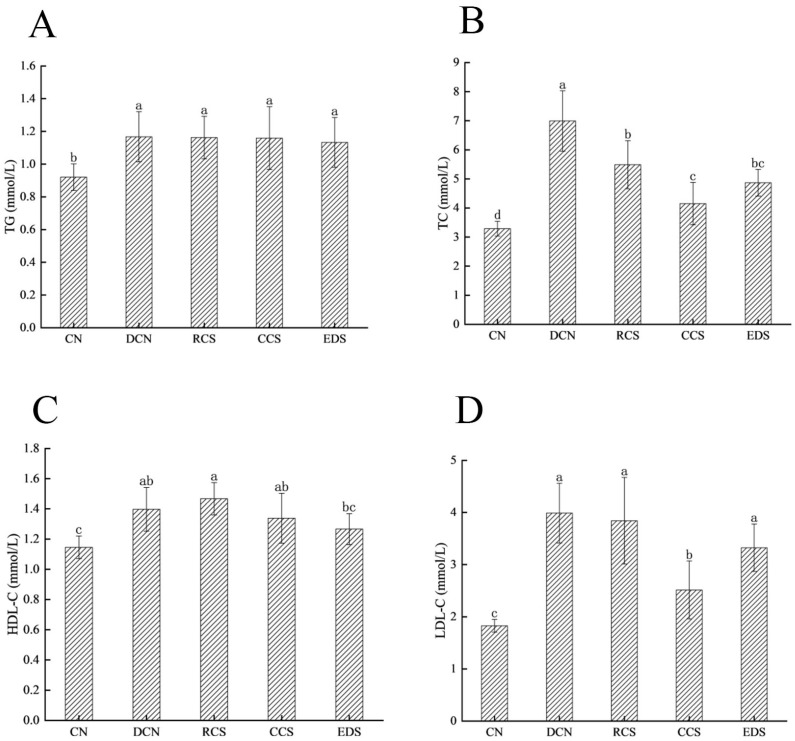
The concentrations of TGs (**A**), TC (**B**), HDL-C (**C**), and LDL-C (**D**) in mice. Each value represents the mean ± SD. Different letters (a–c) indicate significant differences from each other. A statistical analysis was performed using a one-way ANOVA with Tukey’s post hoc test. CN, DCN, RCS, CCS, and EDS denote the normal control group, high-fat and high-sugar control group, chickpea raw starch group, citrate-esterified starch group, and enzymatically debranched starch group, respectively.

**Figure 7 foods-13-01486-f007:**
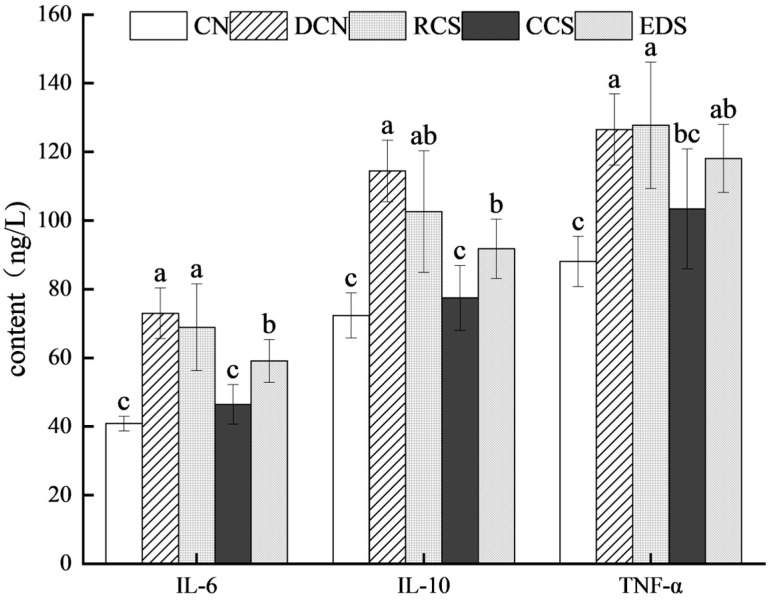
Serum inflammatory factor levels in mice. Each value represents the mean ± SD. Different letters (a–c) indicate significant differences from each other. Statistical analysis was performed using one-way ANOVA with Tukey’s post hoc test. CN, DCN, RCS, CCS, and EDS denote normal control group, high-fat and high-sugar control group, chickpea raw starch group, citrate-esterified starch group, and enzymatically debranched starch group, respectively.

**Figure 8 foods-13-01486-f008:**
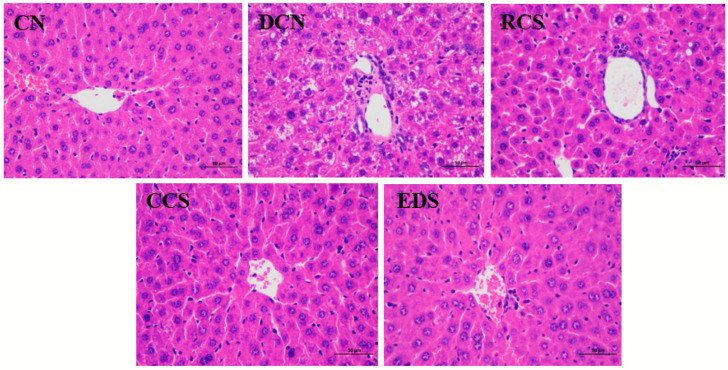
Histopathological examination of liver tissue from mice in different groups (400× magnification). CN: blank control group; DCN: diabetic control group; RCS: chickpea protostarch group; CCS: citrate-esterified starch group; EDS: enzymatically debranched starch group.

**Figure 9 foods-13-01486-f009:**
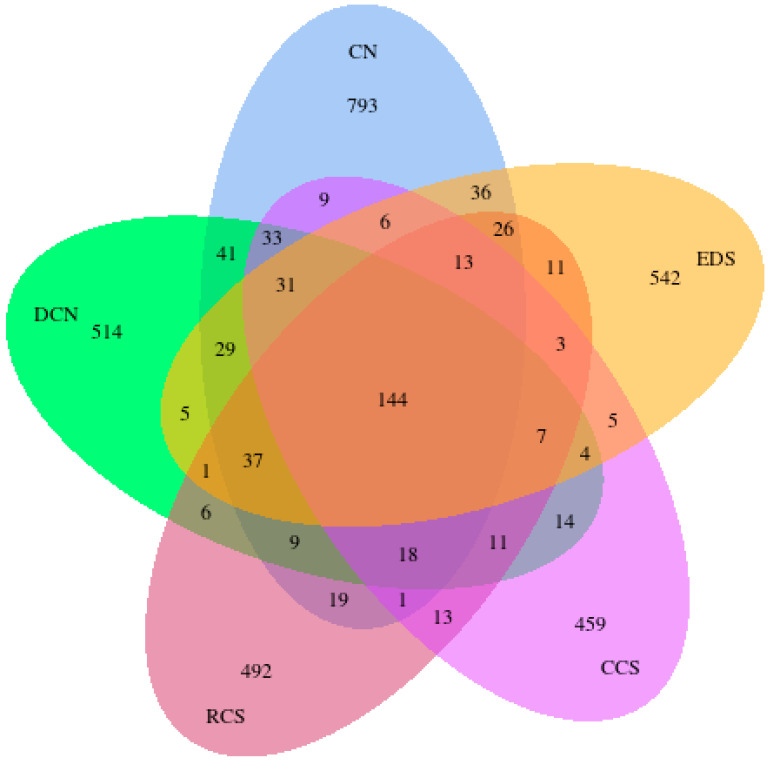
Sample Venn diagram. CN, DCN, RCS, CCS, and EDS denote normal control group, high-fat and high-sugar control group, chickpea raw starch group, citrate-esterified starch group, and enzymatically debranched starch group, respectively.

**Figure 10 foods-13-01486-f010:**
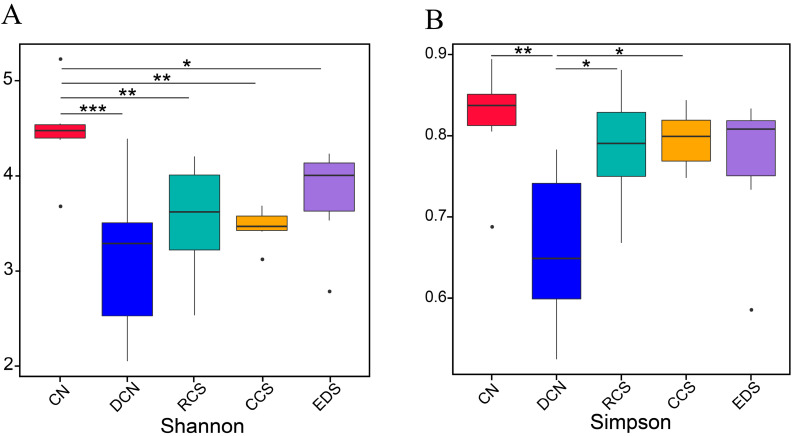
An alpha diversity analysis indicated by the Shannon index (**A**) and Simpson index (**B**). * *p* < 0.05; ** *p* < 0.01, *** *p* < 0.001. CN, DCN, RCS, CCS, and EDS denote the normal control group, high-fat and high-sugar control group, chickpea raw starch group, citrate-esterified starch group, and enzymatically debranched starch group, respectively.

**Figure 11 foods-13-01486-f011:**
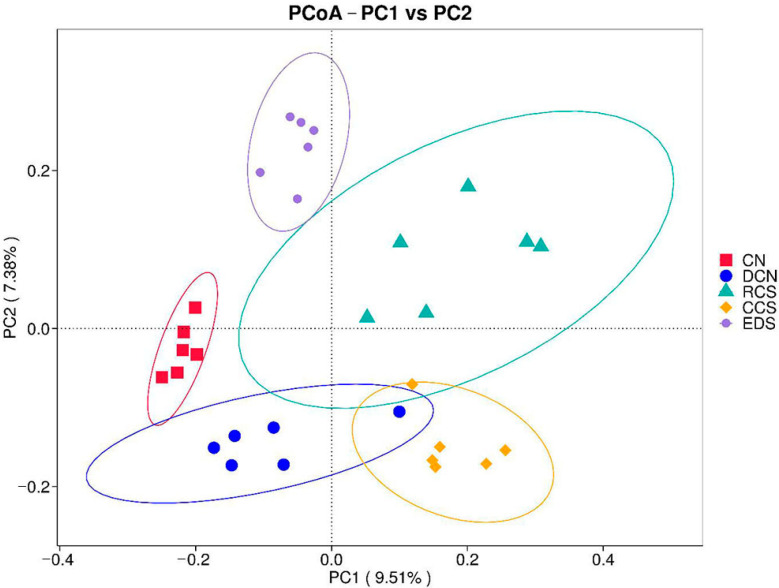
PCoA based on Jaccard. CN, DCN, RCS, CCS, and EDS denote normal control group, high-fat and high-sugar control group, chickpea raw starch group, citrate-esterified starch group, and enzymatically debranched starch group, respectively.

**Figure 12 foods-13-01486-f012:**
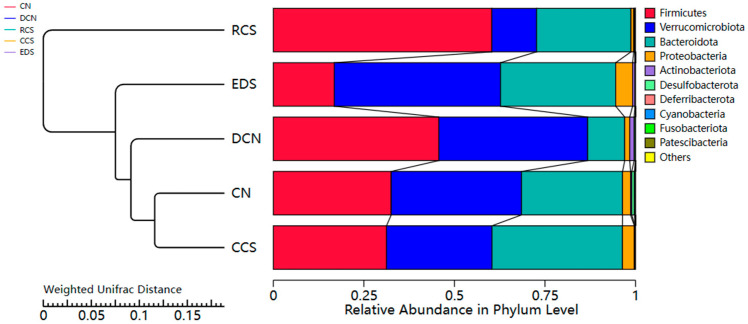
UPGMA clustering tree based on weighted UniFrac distances. CN, DCN, RCS, CCS, and EDS denote normal control group, high-fat and high-sugar control group, chickpea raw starch group, citrate-esterified starch group, and enzymatically debranched starch group, respectively.

**Figure 13 foods-13-01486-f013:**
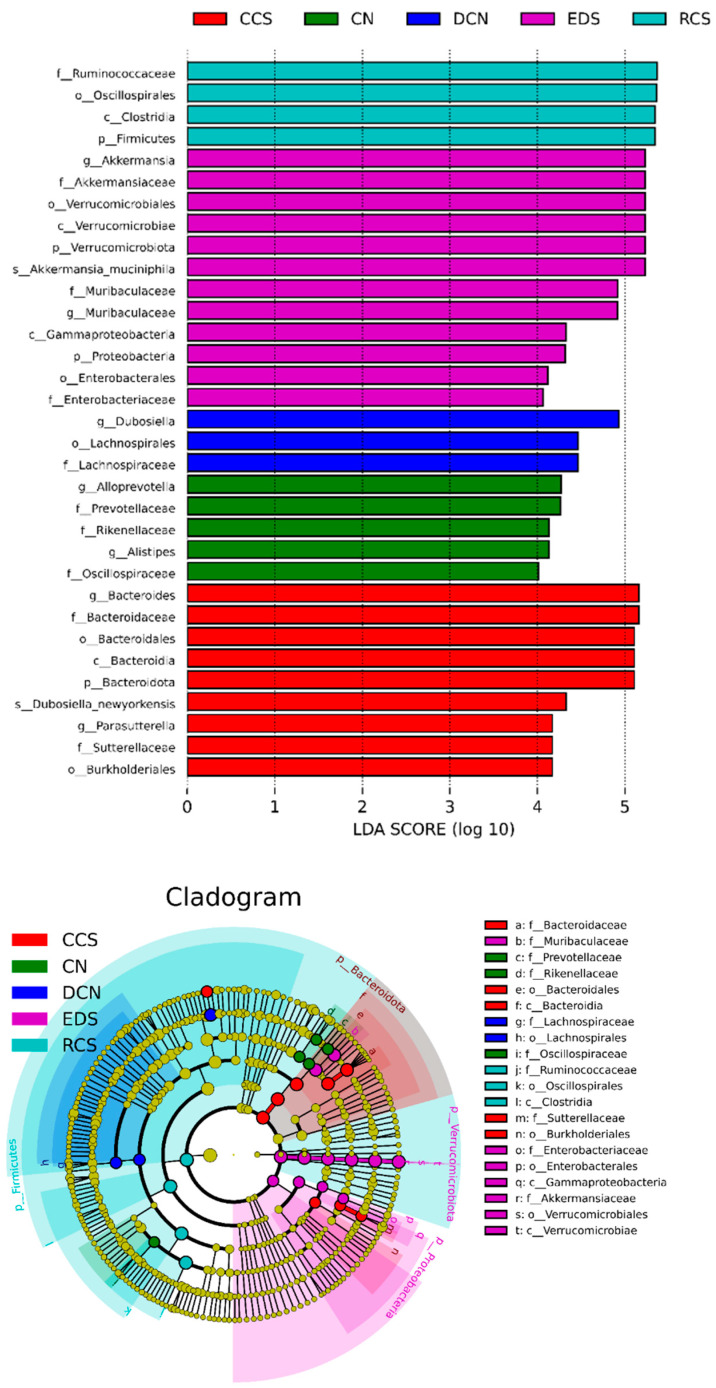
Histogram of species LDA distribution and evolutionary branching. CN, DCN, RCS, CCS, and EDS denote normal control group, high-fat and high-sugar control group, chickpea raw starch group, citrate-esterified starch group, and enzymatically debranched starch group, respectively.

**Figure 14 foods-13-01486-f014:**
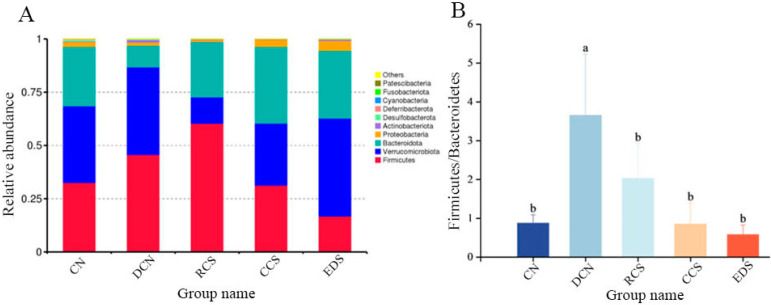
Composition of intestinal flora at phylum level in different groups of mice. (**A**) Histogram of species richness for each group of phylum-level compositions. (**B**) F/B ratio for each group. Each value represents mean ± SD. Different letters (a,b) indicate significant differences from each other. Statistical analysis was performed using one-way ANOVA with Tukey’s post hoc test. CN, DCN, RCS, CCS, and EDS denote normal control group, high-fat and high-sugar control group, chickpea raw starch group, citrate-esterified starch group, and enzymatically debranched starch group, respectively.

**Figure 15 foods-13-01486-f015:**
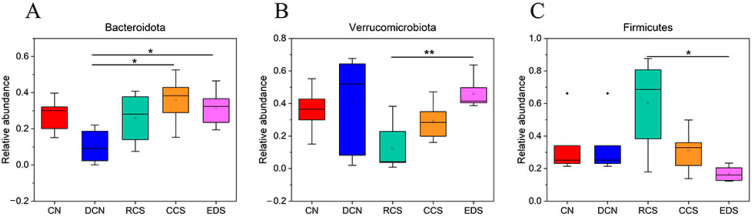
Statistical plot of significant differences in species among groups. (**A**) *Bacteroidota.* (**B**) *Verrucomicrobiota*. (**C**) *Firmicutes*. * *p* < 0.05; ** *p* < 0.01. CN, DCN, RCS, CCS, and EDS denote normal control group, high-fat and high-sugar control group, chickpea raw starch group, citrate-esterified starch group, and enzymatically debranched starch group, respectively.

**Figure 16 foods-13-01486-f016:**
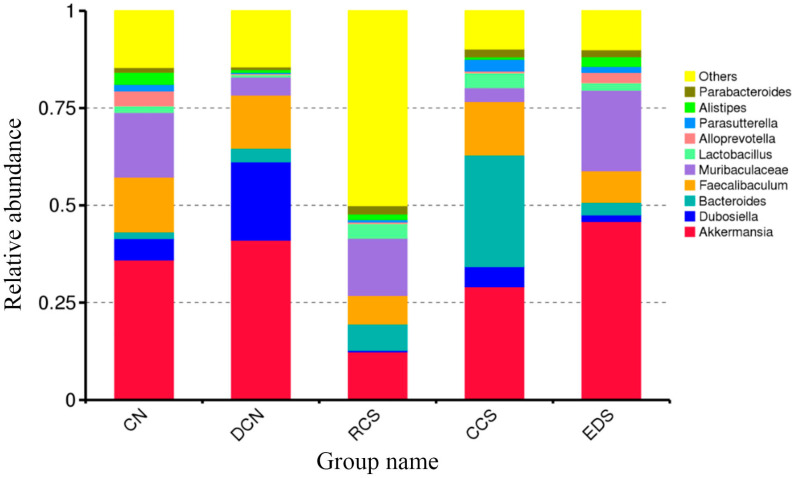
A histogram of relative species richness at the genus level. CN, DCN, RCS, CCS, and EDS denote the normal control group, high-fat and high-sugar control group, chickpea raw starch group, citrate-esterified starch group, and enzymatically debranched starch group, respectively.

**Figure 17 foods-13-01486-f017:**
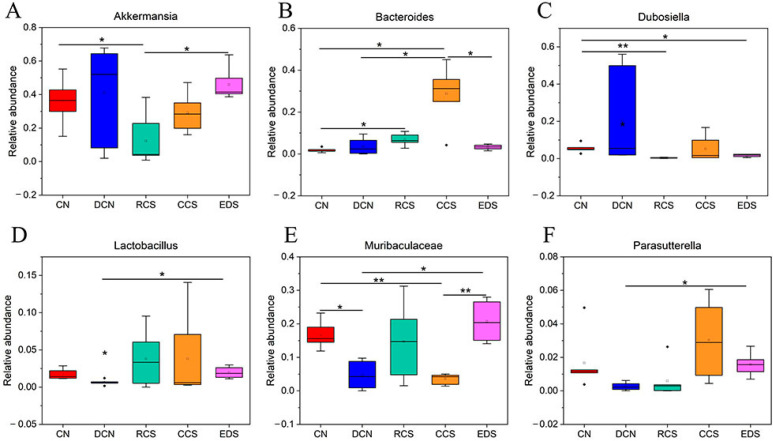
Statistical plots of significant differences in species among groups. (**A**) *Akkermansia*. (**B**) *Bacteroides*. (**C**) *Dubosiella*. (**D**) *Lactobacillus*. (**E**) *Muribaculaceae*. (**F**) *Parasutterella*. * *p* < 0.05; ** *p* < 0.01. CN, DCN, RCS, CCS, and EDS denote normal control group, high-fat and high-sugar control group, chickpea raw starch group, citrate-esterified starch group, and enzymatically debranched starch group, respectively.

**Table 1 foods-13-01486-t001:** Diet compositions of normal chow, starch-supplemented chow, and high-fat and high-sugar chow.

	Basic Feed (g/kg)	Diabetes Control Group (g/kg)	Chickpea Starch Group (g/kg)	Citrate-Esterified Starch Group (CCS Group, g/kg)	Enzymatically Debranched Starch Group (EDS Group, g/kg)
Chickpea starch powder	0	0	234.5	0	0
Citrate-esterified starch (CCS)	0	0	0	234.5	0
Enzymatically debranched starch (EDS)	0	0	0	0	234.5
Corn starch	397.5	234.5	0	0	0
Casein	200	118	118	118	118
Dextrin	132	77.88	77.88	77.88	77.88
Soybean oil	70	41.3	41.3	41.3	41.3
Sucrose	100	59	59	59	59
Microcrystalline cellulose	50	29.5	29.5	29.5	29.5
Cysteine	3	1.77	1.77	1.77	1.77
TBHQ	0.014	0.083	0.083	0.083	0.083
CMC	2.5	1.48	1.48	1.48	1.48
Mixed dimension	10	5.9	5.9	5.9	5.9
Mixed ore	35	20.65	20.65	20.65	20.65
Egg yolk	0	30	30	30	30
Lard	0	180	180	180	180
Sucrose	0	200	200	200	200

**Table 2 foods-13-01486-t002:** Comparison of RS contents of chickpea prepared using different methods *.

Starch	RDS (%)	SDS (%)	RS (%)	1047/1022 cm^−1^
RCS	83.31 ± 1.23 ^a^	9.12 ± 1.71 ^a^	7.57 ± 0.58 ^c^	0.69 ± 0.006 ^b^
EDS	58.06 ± 4.22 ^b^	2.53 ± 1.18 ^b^	38.87 ± 3.05 ^b^	0.73 ± 0.021 ^a^
CCSb	25.16 ± 1.95 ^c^	0.92 ± 0.26 ^b^	74.18 ± 1.74 ^a^	0.59 ± 0.015 ^c^

* RCS, EDS, and CCS denote chickpea starch, enzymatically debranched starch, and citrate-esterified starch, respectively. The percentages of resistant starch (RS), slowly digestible starch (SDS), and rapidly digestible starch (RDS) were determined. Different letters in the same column indicate a significant difference (*p* < 0.05).

## Data Availability

The data used to support the findings of this study are available upon request from the corresponding author. The data are not publicly available due to privacy constraints.

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
