# Peer review of "Characteristics of Citrate-Esterified Starch and Enzymatically Debranched Starch and Their Effects on Diabetic Mice"

_foods, 2024, doi:10.3390/foods13101486_

Round 1

Reviewer 1 Report

Comments and Suggestions for Authors

Manuscript (foods-2954570 ) submitted for review entitled: "Characteristics of citrate esterified starch and enzymatically debranched starch and their effects on diabetic mice" was prepared very carefully and is based on multi-faceted research, properly conducted studies.

The authors characterized modified chickpea resistant starch and evaluated its effect on glucose and lipid parameters, inflammatory markers, and gut microbiota.

The comprehensiveness of the research results on display is worthy of emphasis.

In general, a very well-prepared manuscript. The title of the paper reflects the content contained therein. The purpose of the study has been clearly defined. The structure of the paper is appropriate.

The methods used do not raise any objections and are sufficiently well described. The results obtained were clearly described and discussed. References are appropriately selected thematic and current.

My minor remarks on the reviewed manuscript are as follows:

 I would suggest including the abbreviations of the experimental groups in Table 1 and completing the units for individual dietary components (page 3).

Section 2.14 Statistical Analysis should include other analyses performed, such as PCoA etc.

Page 7, line 250. The data provided is inconsistent with the data in Table 2. Should be : After enzymatic debranching (EDS), the ration of 1047/1022 cm-1 decreased to 0.59. After citric acid estrification (CCS) the ratio increased to 0.73.

Page 9 lines 288-289: The sentence should be completed " with no significant difference between diabetic mice".

Page 11, lines 354-355. Only CCS, and EDS decreased the levels of IL-6 and IL-10 in diabetic mice. RCS should be removed from this sentence.

In Figure 10 (A,B), the horizontal axis should be labeled as the Shannon index and the Simpson index, respectively.

Statistical plots from Figures 10, 15, 17 are difficult to read, their readability should be improved.

Author Response

Ms. Ref. No.:Foods-2954570

Title: Characteristics of citrate-esterified starch and enzymatically debranched starch and their effects on diabetic mice

Journal: Foods

Subject: Response of the revised Foods-2954570

Dear Prof. Dong,

Thank you very much for your valuable comments and suggestions on our manuscript entitled “Characteristics of citrate-esterified starch and enzymatically debranched starch and their effects on diabetic mice (Foods-2954570)”. We have revised the manuscript according to the suggestions. Revisions in the text are marked by red. Thus, the revised version is submitted online for you to consider for publication in the Foods-2954570.

The response to the reviewer’s comments in the manuscript are listed below point by point:

Review from reviewer #1

Comments and Suggestions for Authors

Manuscript (Foods-2954570) submitted for review entitled: "Characteristics of citrate-esterified starch and enzymatically debranched starch and their effects on diabetic mice" was prepared very carefully and is based on multi-faceted research, properly conducted studies.

The authors characterized modified chickpea resistant starch and evaluated its effect on glucose and lipid parameters, inflammatory markers, and gut microbiota.

The comprehensiveness of the research results on display is worthy of emphasis.

In general, a very well-prepared manuscript. The title of the paper reflects the content contained therein. The purpose of the study has been clearly defined. The structure of the paper is appropriate.

The methods used do not raise any objections and are sufficiently well described. The results obtained were clearly described and discussed. References are appropriately selected thematic and current.

My minor remarks on the reviewed manuscript are as follows:

  1. I would suggest including the abbreviations of the experimental groups in Table 1 and completing the units for individual dietary components (page 3).

Response: According to the suggestion of reviewer, we have added the abbreviations of the experimental groups in Table 1 and completed the units for individual dietary components as follows.

Table 1. Diet composition of normal chow, starch-supplemented chow, and high-fat and high-sugar chow

Basic feed (g/kg)

Diabetes control group (g/kg)

Chickpea starch group (g/kg)

Citrate-esterified starch group (CCS group, g/kg)

Enzymatically debranched starch group (EDS group, g/kg)

Chickpea starch powder

0

0

234.5

0

0

Citrate-esterified starch (CCS)

0

0

0

234.5

0

Enzymatically debranched starch (EDS)

0

0

0

0

234.5

Corn Starch

397.5

234.5

0

0

0

Casein

200

118

118

118

118

Dextrin

132

77.88

77.88

77.88

77.88

Soybean Oil

70

41.3

41.3

41.3

41.3

Sucrose

100

59

59

59

59

Microcrystalline cellulose

50

29.5

29.5

29.5

29.5

Cysteine

3

1.77

1.77

1.77

1.77

TBHQ

0.014

0.083

0.083

0.083

0.083

CMC

2.5

1.48

1.48

1.48

1.48

Mixed dimension

10

5.9

5.9

5.9

5.9

Mixed ore

35

20.65

20.65

20.65

20.65

Egg yolk

0

30

30

30

30

Lard

0

180

180

180

180

Sucrose

0

200

200

200

200

  1. Section 2.14 Statistical Analysis should include other analyses performed, such as PCoA etc.

Response: According to the suggestion of reviewer, we have added other analyses performed in Section 2.14 as follows.

A Venn diagram can be used to analyse common and unique feature sequences in sample groups. High-quality sequences were categorized into operational taxonomic units (OTUs) with a similarity truncation value of 98%. Alpha-diversity was used to analyze the diversity of microbial communities in the samples. Alpha-diversity indices, including Shannon and Simpson [22], were calculated using Quantitative Insights into Microbial Ecology (QIIME). The beta- diversity among different groups was analyzed by principal coordinate analysis (PCoA). PCoA results were displayed using ade4 package and ggplot2 package in R software (Version 2.15.3). The unweighted pair-group method with arithmetic mean analysis (UPGMA) is a common clustering analysis method used to solve classification problems. This method involves constructing a clustering tree to analyse the similarity among sample groups. Linear discriminant analysis (LDA) effect size (LEfSe) was used to find the biomarkers with statistical differences between groups, which was presented by Histogram of Linear Discriminant Analysis distribution and cladogram [23]. The LEfSe software (Version 1.0) was used to do LEfSe analysis (LDA score threshold: 4) so as to find out the biomarkers.

  1. Hugerth, L.W.; Andersson, A.F. Analysing microbial community composition through amplicon sequencing: from sampling to hypothesis testing. Front. Microbiol.2017, 8, 1561.
  2. Segata, N.; Izard, J.; Waldron, L.; Gevers, D.; Miropolsky, L.; Garrett, W.S.; Huttenhower, C. Metagenomic biomarker discovery and explanation. Genome Biol.2011, 12(6), R60.
  3. Page 7, line 250. The data provided is inconsistent with the data in Table 2. Should be : After enzymatic debranching (EDS), the ration of 1047/1022 cm-1 decreased to 0.59. After citric acid estrification (CCS) the ratio increased to 0.73.

3. Page 7, line 250. The data provided is inconsistent with the data in Table 2. Should be : After enzymatic debranching (EDS), the ration of 1047/1022 cm-1 decreased to 0.59. After citric acid estrification (CCS) the ratio increased to 0.73.

Response: Thanks a lot for the reviewer’s suggestion. After careful checking, there were some errors in Table 2. The data provided in line 250 was right. We have corrected the data in Table 2. We also have checked the whole manuscript.

Table 2 Comparison of the RS contents of chickpea prepared by different methods*

Starch

RDS (%)

SDS (%)

RS (%)

1047/1022 cm-1

RCS

83.31±1.23a

9.12±1.71a

7.57±0.58c

0.69±0.006b

EDS

58.06±4.22b

2.53±1.18b

38.87±3.05b

0.73±0.021a

CCS

25.16±1.95c

0.92±0.26b

74.18±1.74a

0.59±0.015c

*RCS, EDS, and CCS denote chickpea starch, enzymatically debranched starch, and citrate-esterified starch, respectively. The percentages of resistant starch (RS), slowly digestible starch (SDS) and rapidly digestible starch (RDS) were determined. Different letters in the same column indicate a significant difference (P<0.05).

  1. Page 9 lines 288-289: The sentence should be completed " with no significant difference between diabetic mice".

Response: According to the suggestion of reviewer, we have completed the sentence as follows.

Before intervention, the blood glucose levels of the mice in the model group and the intervention group ranged from 23-25 mmol/L, with no significant difference between diabetic mice (Figure 5).

  1. Page 11, lines 354-355. Only CCS, and EDS decreased the levels of IL-6 and IL-10 in diabetic mice. RCS should be removed from this sentence.

Response: According to the suggestion of reviewer, we have removed RCS from this sentence as follows.

Thus, CCS and EDS can decrease the levels of IL-6 and IL-10 in diabetic mice.

  1. In Figure 10 (A,B), the horizontal axis should be labeled as the Shannon index and the Simpson index, respectively.

Response: According to the suggestion of reviewer, we have labeled the horizontal axis in figures 10 (A,B) as the Shannon index and the Simpson index, respectively.

Figure 10. Alpha diversity analysis indicated by the Shannon index (A) and Simpson index (B). * P<0.05, ** P<0.01. CN, DCN, RCS, CCS, EDS denote the normal control group, high-fat and high-sugar control group, chickpea raw starch group, citrate-esterified starch group, and enzymatically debranched starch group, respectively.

7.Statistical plots from Figures 10, 15, 17 are difficult to read, their readability should be improved.

Response: According to the suggestion of reviewer, we have improved the readability of figures 10, 15 and 17 as follows.

Figure 10. Alpha diversity analysis indicated by the Shannon index (A) and Simpson index (B). * P<0.05, ** P<0.01. CN, DCN, RCS, CCS, EDS denote the normal control group, high-fat and high-sugar control group, chickpea raw starch group, citrate-esterified starch group, and enzymatically debranched starch group, respectively.

Figure 15. Statistical plot of significant differences in species among groups. (A) Bacteroidota. (B) Verrucomicrobiota. (C) Firmicutes. * P<0.05, ** P<0.01. CN, DCN, RCS, CCS, EDS denote the normal control group, high-fat and high-sugar control group, chickpea raw starch group, citrate-esterified starch group, and enzymatically debranched starch group, respectively.

Figure 17. Statistical plots of significant differences in species among groups. (A) Akkermansia. (B) Bacteroides. (C) Dubosiella. (D) Lactobacillus. (E) Muribaculaceae. (F) Parasutterella. * P<0.05, ** P<0.01. CN, DCN, RCS, CCS, EDS denote the normal control group, high-fat and high-sugar control group, chickpea raw starch group, citrate-esterified starch group, and enzymatically debranched starch group, respectively.

Reviewer 2 Report

Comments and Suggestions for Authors

The manuscript “Characteristics of citrate-esterified starch and enzymatically debranched starch and their effects on diabetic mice” is a relevant article with a well-structured experimental design. Despite previous corrections, some gaps were still found regarding the wording of the introduction and the presentation of methods and results. Below are the points that need to be corrected.

Comments:

Lines 31-33 and 41-42: the authors should add references to this data.

Lines 34-39: the authors present information about chickpeas in the second paragraph. This is a relevant point for the work; therefore, more studies should be added as references, in addition to the only one cited.

Lines 55-56: the authors must cite the studies referring to the excerpt: “Moreover, the yield of RS from chickpeas using this technique is low, and only high-fat mouse models have been studied”.

Lines 308-321: the authors must add more than two references to this paragraph.

Lines 379-444: where are the methodological analyses described, including the test used, type of data, and form of graphical presentation, demonstrated as results in this topic (Venn diagram, alpha diversity analysis, PCoA, UPGMA, LEfSe)?

Lines 379-444: still at this point, the authors should remove this methodological information from the results section and direct it to the methods section:

“A Venn diagram can be used to analyze sequences of common and unique features in groups of samples” (381-382).

“Alpha diversity was used to analyze the diversity of microbial communities in the samples” (396-397).

“UPGMA is a common cluster analysis method used to solve classification problems” (427-428).

Lines 379-444: as a suggestion to the previous point, an interesting description is presented in topic 3.10.3.

Line 549: the authors should add study limitations and future perspectives in the conclusions section.

Author Response

Ms. Ref. No.:Foods-2954570

Title: Characteristics of citrate-esterified starch and enzymatically debranched starch and their effects on diabetic mice

Journal: Foods

Subject: Response of the revised Foods-2954570

Dear Prof. Dong,

Thank you very much for your valuable comments and suggestions on our manuscript entitled “Characteristics of citrate-esterified starch and enzymatically debranched starch and their effects on diabetic mice (Foods-2954570)”. We have revised the manuscript according to the suggestions. Revisions in the text are marked by red. Thus, the revised version is submitted online for you to consider for publication in the Foods-2954570.

The response to the reviewer’s comments in the manuscript are listed below point by point:

Review from reviewer #2 

Comments and Suggestions for Authors

The manuscript “Characteristics of citrate-esterified starch and enzymatically debranched starch and their effects on diabetic mice” is a relevant article with a well-structured experimental design. Despite previous corrections, some gaps were still found regarding the wording of the introduction and the presentation of methods and results. Below are the points that need to be corrected.

Comments:

  1. Lines 31-33 and 41-42: the authors should add references to this data.

Response: According to the suggestion of reviewer, we have added references to the mentioned datas as follows.

Currently, patients with diabetes constitute approximately 10% of the global population and play a significant role in the continuous growth of healthcare system expenditures [5].

RS, a novel dietary fibre, has recently attracted the attention of researchers [9]. RS can stabilize postprandial blood sugar and insulin levels in diabetic patients [10].

5. Collaboration, N.C.D.R.F. Worldwide trends in diabetes since 1980: a pooled analysis of 751 population-based studies with 4.4 million participants. Lancet 2016, 387, 1513-1530.

9. Wang, C.; McClements, D.J.; Jiao,A.; Wang, J.; Jin Z.; Qiu, C. Resistant starch and its nanoparticles: recent advances in their green synthesis and application as functional food ingredients and bioactive delivery systems. Trends Food Sci. Technol.2022, 119, 90-100.

10. Jiali, L.; Wu, Z.; Liu, L.; Yang, J.; Wang, L.; Li, Z.; Liu, L. The research advance of resistant starch: structural characteristics, modification method, immunomodulatory function, and its delivery systems application. Crit. Rev. Food Sci. Nutr.2023, 6, 1-18.

  2. Lines 34-39: the authors present information about chickpeas in the second paragraph. This is a relevant point for the work; therefore, more studies should be added as references, in addition to the only one cited.

Response: According to the suggestion of reviewer, we have added two references to the second paragraph as follows.

Chickpea (Cicer arietinum Linn), which belongs to the legume family, is an annual or perennial herbaceous plant [6]. Chickpea is rich in starch, protein, dietary fibre, flavonoids and vitamins [7].

6. Singh, R.; Sharma, P.; Varshney, R.K.; Sharma, S.K.; Singh, N.K. Chickpea improvement: role of wild species and genetic markers. Biotechnol. Genet. Eng.Rev. 2008, 25,267-314.

7. Kaur, R.; Prasad, K. Technological, processing and nutritional aspects of chickpea (Cicer arietinum) - A review. Trends Food Sci. Technol. 2021, 109, 448-463.

3. Lines 55-56: the authors must cite the studies referring to the excerpt: “Moreover, the yield of RS from chickpeas using this technique is low, and only high-fat mouse models have been studied”.

Response: According to the suggestion of reviewer, we have cited the studies referring to the excerpt: “Moreover, the yield of RS from chickpeas using this technique is low, and only high-fat mouse models have been studied” as follows.

Moreover, the yield of RS from chickpeas using this technique is low, and only high-fat mouse models have been studied [16, 17].

16. Demirkesen-Bicak, H.; Tacer-Caba, Z.; Nilufer-Erdil, D. Pullulanase treatments to increase resistant starch content of black chickpea (Cicer arietinumL.) starch and the effects on starch properties. Int. J. Biol. Macromol. 2018, 111, 505-513.

17. Zhao, M.; Cui, W.; Hu, X.; Ma, Z. Anti-hyperlipidemic and ameliorative effects of chickpea starch and resistant starch in mice with high fat diet induced obesity are associated with their multi-scale structural characteristics.Food Funct.2022, 13, 5135-5152.

4. Lines 308-321: the authors must add more than two references to this paragraph.

Response: According to the suggestion of reviewer, we have added three references to the paragraph as follows.

T2DM is a complex chronic metabolic disease caused by insulin resistance in peripheral tissues [39]. Lipid metabolism can cause insulin resistance [40].There is a close connection between lipid metabolism and glucose metabolism [41]. The inability to maintain lipid levels within the normal range over a long period is a major cause of chronic complications in patients with T2DM. Therefore, the regulation of lipid metabolism is an essential component of RCS modulation. TG, TC, HDL-C, and cholesterol LDL-C reflect the body's lipid metabolism and possess many properties, such as anti-inflammatory, antioxidant, and antiapoptotic effects [42].

39. Zimmet, P.Z.; Magliano, D.J.; Herman, W.H.; Shaw, J.E. Diabetes: a 21st century challenge. Lancet Diabetes Endocrinol.2014, 2(1), 56-64.

41. Saltiel, A.R.; Kahn, C.R. Insulin signalling and the regulation of glucose and lipid metabolism. Nature2001,414, 799-806.

42. Wang, L.; Yan, N.; Zhang, M.; Pan, R.; Dang, Y.; Niu, Y. The association between blood glucose levels and lipids or lipid ratios in type 2 diabetes patients: a cross-sectional study. Front. Endocrinol. 2022, 13, 969080.

5. Lines 379-444: where are the methodological analyses described, including the test used, type of data, and form of graphical presentation, demonstrated as results in this topic (Venn diagram, alpha diversity analysis, PCoA, UPGMA, LEfSe)?

Response: According to the suggestion of reviewer, we have added the methodological analyses described in Section 2.14 as follows.

A Venn diagram can be used to analyse common and unique feature sequences in sample groups. High-quality sequences were categorized into operational taxonomic units (OTUs) with a similarity truncation value of 98%. Alpha-diversity was used to analyze the diversity of microbial communities in the samples. Alpha-diversity indices, including Shannon and Simpson [22], were calculated using Quantitative Insights into Microbial Ecology (QIIME). The beta- diversity among different groups was analyzed by principal coordinate analysis (PCoA). PCoA results were displayed using ade4 package and ggplot2 package in R software (Version 2.15.3). The unweighted pair-group method with arithmetic mean analysis (UPGMA) is a common clustering analysis method used to solve classification problems. This method involves constructing a clustering tree to analyse the similarity among sample groups. Linear discriminant analysis (LDA) effect size (LEfSe) was used to find the biomarkers with statistical differences between groups, which was presented by Histogram of Linear Discriminant Analysis distribution and cladogram [23]. The LEfSe software (Version 1.0) was used to do LEfSe analysis (LDA score threshold: 4) so as to find out the biomarkers.

22. Hugerth, L.W.; Andersson, A.F. Analysing microbial community composition through amplicon sequencing: from sampling to hypothesis testing. Front. Microbiol.2017, 8, 1561.

23. Segata, N.; Izard, J.; Waldron, L.; Gevers, D.; Miropolsky, L.; Garrett, W.S.; Huttenhower, C. Metagenomic biomarker discovery and explanation. Genome Biol.2011, 12(6), R60.

6. Lines 379-444: still at this point, the authors should remove this methodological information from the results section and direct it to the methods section:

“A Venn diagram can be used to analyze sequences of common and unique features in groups of samples” (381-382).

“Alpha diversity was used to analyze the diversity of microbial communities in the samples” (396-397).

“UPGMA is a common cluster analysis method used to solve classification problems” (427-428). 

Lines 379-444: as a suggestion to the previous point, an interesting description is presented in topic 3.10.3.  

Response: According to the suggestion of reviewer, we have removed these methodological information from the results section and directed them to the methods section.

7. Line 549: the authors should add study limitations and future perspectives in the conclusions section.

Response: According to the suggestion of reviewer, we have added the study limitations and future perspectives in the conclusions section as follows.

In the paper, the effect of modified chickpea RS on diabetic mouse model was studied, but the effect on diabetic patients needs further study. In addition, further studies are needed to determine the amount of its addition in staple foods, such as bread, steamed bread and noodles, not only to meet its functional properties such as reducing postprandial blood sugar and improving intestinal flora, but also to consider its impact on the sensory quality of food. It is believed that these studies will provide theoretical guidance for the application of chickpea RS in the development of special food for diabetic population.